# Channel Features and API Frequency-Based Transformer Model for Malware Identification

**DOI:** 10.3390/s24020580

**Published:** 2024-01-17

**Authors:** Liping Qian, Lin Cong

**Affiliations:** School of Electrical and Information Engineering, Beijing University of Civil Engineering and Architecture, Beijing 100044, China; qianliping@bucea.edu.cn

**Keywords:** malware identification, deep learning, dynamic analysis, API sequence, transformer

## Abstract

Malicious software (malware), in various forms and variants, continues to pose significant threats to user information security. Researchers have identified the effectiveness of utilizing API call sequences to identify malware. However, the evasion techniques employed by malware, such as obfuscation and complex API call sequences, challenge existing detection methods. This research addresses this issue by introducing CAFTrans, a novel transformer-based model for malware detection. We enhance the traditional transformer encoder with a one-dimensional channel attention module (1D-CAM) to improve the correlation between API call vector features, thereby enhancing feature embedding. A word frequency reinforcement module is also implemented to refine API features by preserving low-frequency API features. To capture subtle relationships between APIs and achieve more accurate identification of features for different types of malware, we leverage convolutional neural networks (CNNs) and long short-term memory (LSTM) networks. Experimental results demonstrate the effectiveness of CAFTrans, achieving state-of-the-art performance on the mal-api-2019 dataset with an F1 score of 0.65252 and an AUC of 0.8913. The findings suggest that CAFTrans improves accuracy in distinguishing between various types of malware and exhibits enhanced recognition capabilities for unknown samples and adversarial attacks.

## 1. Introduction

With the continuous development of Internet technology, malware attacks are occurring frequently, leading to an ongoing arms race between malware authors and security software companies. According to AV-TEST reports, there has been a significant surge in the number of discovered malware samples in recent years. From 2019 to 2020, the addition rose to 113 million, and from 2020 to 2021, it further increased to over 170 million, representing a growth of nearly 33%, making it one of the most robust figures in the past decade [1]. Currently, the phenomenon of cyber attacks is on the rise, with many servers facing the pressure of DDoS attacks, and ransomware and phishing emails are constantly emerging [2]. In order to safeguard the security of user information, accurately detecting different types of malware in the system has become a challenging issue in the field of security [3].

Malware can be defined as code that operates without being detected by system administrators and executes malicious actions through system vulnerabilities, thereby affecting the system’s regular operation [4]. This insidious software is designed with nefarious intent, seeking unauthorized access to computer systems and network resources. The realm of malware encompasses a wide array of variants, each with distinct functionalities, including backdoors, botnets, downloaders, spyware, rootkits, ransomware, adware, worms, viruses, and more, each with its pernicious purpose [5].

Malware analysis methods can be categorized into static analysis and dynamic analysis based on whether they execute malicious code. Early researchers favored signature-based static analysis methods for feature extraction [6,7,8]. Specifically, signature-based methods distinguish different types of malware by identifying characteristic segments in the source code binary, requiring significant manual effort from skilled researchers. As detection techniques evolve, malware creators continually enhance the architecture of malicious code by inserting irrelevant code logic or employing techniques such as polymorphism (encryption/packing) and dynamic execution of malicious code to conceal malicious behavior [4]. Due to their limitation of recognizing only existing features, signature-based methods fail to detect malicious intent in the face of adversarial attacks. This also results in the vulnerability of feature extraction methods in static analysis, making it easier for attackers to employ obfuscation techniques to evade detection. This significantly restricts the capability to detect novel zero-day attacks. Currently, researchers are actively seeking an automated and high-precision detection technology.

Dynamic analysis involves executing malware in a real-time environment or a virtual environment like a sandbox to extract features, compensating to some extent for the shortcomings of static analysis. We can obtain behavioral information about the software in a virtual environment through dynamic analysis, such as network activities, registry interactions, DLL loading, etc. [3]. API call features play a crucial role in dynamic analysis. These calls contain information about the software’s utilization and modification of system resources. When individual API calls are combined into a sequence, it signifies that the software has executed a series of coherent operations on the system, and there are unique semantic associations among the APIs in the sequence. Capturing the calling relationships in the API sequence of malware enables better differentiation of different types of malware, which is highly significant [9].

The API sequences of regular software and malware are constructed with specific differences stemming from variations in their functionalities. Malware explores vulnerabilities or executes malicious activities by systematically traversing system API calls to circumvent permission controls. Additionally, malware may attempt to communicate with a malicious control server, transmit or receive malicious instructions, or transfer sensitive information without user consent. More severe forms of malware may endeavor to modify system configurations, delete or tamper with files, or perform other operations that threaten system security. Different types of malware involve distinct API commands, reflecting their diverse functional requirements. This functionality-driven distinction manifests in the API sequences, aiding malware detection systems in identifying potential threats.

In recent years, with the advancement of deep learning technologies, researchers have begun employing deep neural networks for malware detection. Applications such as convolutional neural networks (CNNs), recurrent neural networks (RNNs), and large pre-trained models from natural language processing have been applied to the task of detecting malware API sequences [10,11,12,13]. Although researchers have achieved excellent results using API sequence features in malware detection tasks, there are still research gaps in classifying different types of malware and detecting unknown attacks. Detecting unknown attacks has become particularly crucial, categorizing them into two types: originating from the same family and attacks of unknown types. Presently, researchers are predominantly focused on classifying known data, yet we frequently encounter entirely new unknown samples in the real world. This experiment addresses certain limitations in current research by extracting malicious behavior characteristics from known data to detect unknown samples. Therefore, extracting features from various types of malware API sequences and capturing semantic information to enhance the identification of unknown samples is essential.

Text classification is a significant problem in natural language processing (NLP), where the task involves learning from text and extracting features to enable automatic categorization. Notably, API sequences can also be regarded as a specific type of text. In this context, extracting semantic relationships from API sequences aids in better identifying the distinct functionalities executed by various types of malicious software, akin to perceiving potential sentiments. This approach effectively enhances accuracy and performance in our classification tasks.

We propose a malware detection model, CAFTrans, based on API sequences. The model enhances the representation of API features by utilizing natural language processing techniques combined with call frequency and attention to unique API behaviors. CAFTrans has two distinctive features: a unique one-dimensional channel attention module and a feature reinforcement module based on word frequency. Firstly, API sequences undergo embedding representation through a transformer network. In each encoder, we employ a one-dimensional channel attention module to reinforce the specificity of features, making API vector features more distinct. Subsequently, a word frequency reinforcement module is applied to enhance the feature values of high-frequency API calls while preserving the features of low-frequency API calls. Finally, a downstream classifier composed of CNN and LSTM is used to identify the unique correlations in API sequences of different malware types, enabling better differentiation among various types of malware. The contributions of this paper are as follows:We integrate NLP techniques into the field of computer security and have designed a novel transformer model specifically for detecting malware API sequences;By introducing a unique one-dimensional channel attention mechanism, we enhance the model’s effectiveness in API embedding. Additionally, the word frequency reinforcement module not only preserves the details of the sequences but also emphasizes the features of segments, enabling the model to better distinguish between different types of malware. We showcase the model’s classification results by visualizing clustering effects. Furthermore, we conducted ablation experiments to validate the contributions of each module to the detection performance;We subjected the model to adversarial testing using unknown samples. The test results indicate that the model demonstrates increased robustness, effectively identifying unknown samples.We compared our model with previous researchers’ work and various machine learning methods to validate its performance. Our model achieved superior results, with a mean time to detection (MTTD) of 1.2 ms for individual samples.

This paper’s remaining content is structured as follows: Section 2 discusses previous researchers’ relevant work and background information on malware APIs. Section 3 provides a detailed overview of the design of our model. Following that, Section 4 offers the experimental details, evaluating the performance of our proposed model for malware detection using the mal-api-2019 dataset. The experiments encompass analyses of detection performance, ablation experiments, data cluster analysis, and detection of attacks with unknown samples. In Section 5, we describe some limitations and provide discussions. Finally, Section 6 summarizes the paper.

## 2. Related Work

This section presents an extensive review of contemporary research that utilizes API sequences as features in dynamic analysis. Furthermore, we explore the current state-of-the-art obfuscation techniques adopted by malware creators.

Researchers widely recognize API sequences as pivotal features for malware detection. Various approaches have been extensively employed in this domain, including frequency-based, sequence-based, and graph-based methods.

### 2.1. Frequency-Based Methods

Tian et al. [14] employed pattern recognition and statistical frequency methods to analyze API sequences. They achieved an accuracy of over 97% in distinguishing between malicious and benign software. The authors’ work demonstrates that studying the frequency of API calls in malware can provide theoretical support for identifying its malicious nature.

Kim [15] was inspired by NLP and proposed a novel dynamic analysis method for detecting malware. Kim traced the API call characteristics of the malware and applied n-grams for feature extraction from the sequences. Kim calculated each gram’s term frequency-inverse document frequency (TF-IDF) weight. Finally, Kim performed binary classification on weighted n-gram features using an SVM classifier. The author achieved a detection accuracy of 96% on the dataset collected from VirusShare.

Dabas et al. [16] collected three feature sets, which included API call usage, API call frequency, and API call sequences. These feature sets were augmented using TF-IDF, resulting in a more robust integrated API feature set. Experimental results demonstrated that when tested with SVM, KNN, LR, and DT algorithms, all achieved accuracy rates of over 99.6%. By utilizing TF-IDF to assist the feature set, they effectively address the problem of high dimensionality in the integrated API feature set.

### 2.2. Sequence Association-Based Methods

With the advancement of NLP techniques, many researchers have started to employ transfer learning methods to apply NLP techniques to malware detection tasks. Recurrent neural networks (RNNs) and convolutional neural networks (CNNs), based on NLP, have been widely utilized in API sequence classification tasks to learn the malicious semantics within them.

Zhang et al. [17] proposed an innovative method to detect zero-day and obfuscated malware. This approach utilizes feature hashing techniques to encode API names and their corresponding call parameters. It employs multiple 1D convolutional networks to transform the features extracted from each API call. Finally, a Bi-LSTM is used to learn both forward and backward semantic information from the transformed features. Through experimentation, the researchers demonstrated that their model outperformed baseline models on the extracted dataset, showing superior performance.

Li et al. [18] devised a novel architecture for extracting API feature associations: API semantic chains and API phrases. In API semantic chains, the authors represented the semantic information of each API call as a four-dimensional vector with behavior, action objects, categories, and attributes. They then used a 1D convolutional network to merge these features. The authors designed a multi-scale convolutional structure for API phrases to extract association information at different distances within the sequence. The accuracy of the data extracted by the authors reached 97.31%, and the F1 score was 0.9724.

### 2.3. Graph-Based Methods

In their work, Amer et al. [19] established separate behavior transition graph models for normal software and malware by exploring the contextual relationships within API scenes. Additionally, they proposed a novel heuristic detection algorithm that determines the maliciousness of a target by calculating its confidence in malicious transitions. This method proved effective in detecting obfuscated API sequences. The proposed model demonstrated its ability to effectively learn both explicit and implicit relationships within API sequences. The results showed that the model could accurately detect and analyze the presence of malicious behaviors in the API sequences.

Li et al. [20] introduced a novel malware analysis framework called DMalNet. This framework utilizes a novel encoder architecture to combine API names with their corresponding parameter features and extract semantic features from them. These semantic feature relationships are then converted into graph-based structural representations. Finally, an enhanced graph isomorphism network (GINE) and a graph attention network (GAT) are employed to learn the features of the API call graph. DMalNet demonstrated excellent performance in both malware detection and classification tasks.

Chen et al. [21] proposed a deep neural network (DNN)-based method for malware detection called MalPro. MalPro utilizes a logic-regression-based approach to calculate a weight value representing the sensitivity of API pairs to malicious behavior. It then generates a process graph by processing the API sequences and uses fully connected layers to output the graph’s features. Finally, attention weights and process graph features are combined using weights and learned through a DNN to extract essential behavioral information.

However, to ensure the integrity of connections between nodes in the graph, it is often necessary to handle longer raw API sequence inputs to maintain the accuracy of the analysis process. Nevertheless, this processing requirement may result in a need to wait for the completion of API extraction work before conducting malicious checks, contradicting the principles of real-time detection in dynamic analysis. Additionally, due to the unique structure of the graph involving a substantial amount of edge and node information, the analysis process may incur higher computational costs to efficiently address the demands of handling nodes and edges in the graph.

### 2.4. Adversarial Attack Trouble

With researchers increasingly turning to deep learning techniques for malware detection, the challenge of countering adversarial attacks using obfuscation methods [22] has become a classic issue, as observed in other domains. Szegedy et al. [23] were among the first to identify vulnerabilities in deep learning networks, successfully achieving misclassification of images by inserting imperceptible pixels. Subsequently, adversarial attack techniques swiftly expanded to natural language processing, biometrics, malware detection, and other domains.

Malware authors often employ obfuscation techniques such as packing and compression, dynamic code generation, control flow obfuscation (encryption, oligomorphic, metamorphic, stealth) [24], and packaging. However, these obfuscation methods do not yield satisfactory results when applied to detection based on API sequences. This is because API sequence features are extracted by simulating malware execution. Regardless of how malware authors modify code structures, the presented malicious behavior remains unchanged, i.e., API sequence features remain constant. Directly obfuscating API sequences would jeopardize the normal execution of the original program. Researchers have been dedicated to exploring methods that ensure the regular operation of malware while bypassing obfuscation in API sequence-based detection methods. Inserting no-operation (NOP) instructions is common in this context.

Gibert et al. [25] introduced an obfuscation technique capable of evading detection by convolutional neural networks (CNNs). By strategically inserting NOP instructions into the malicious PE files, they successfully led the classifier to misclassify the malware. Their experiments showed a remarkable 56.53% decrease in detection accuracy for a particular family of malware samples, achieving a 100% success rate in obfuscation.

Park et al. [26] introduced the Adversarial Malware Alignment Obfuscation (AMAO) algorithm, which calculates the optimal positions for inserting semantic NOP instructions or sequences that do not affect the program logic. The authors utilized the FGSM or C&W algorithms to generate adversarial samples. During the evaluation, if the obfuscated sample failed to evade classifier detection, the adversarial attack generation process was iteratively repeated until successful obfuscation was achieved. Their experiments effectively reduced the accuracy of CNNs to 0%.

The researchers presented a method for obfuscating API sequences, which effectively evades detection by RNNs [27]. They utilized a generative RNN to process the original API sequences and generate adversarial API sequences. Experimental results demonstrated that this obfuscation technique can successfully evade detection by various structures of recurrent neural networks.

Drawing from the insights provided by the researchers’ work above, it becomes evident that adversarial samples pose a critical challenge in malware detection. Consequently, designing a more robust network architecture to detect adversarial samples effectively is essential.

## 3. Proposed Method

In this section, we present our transformer-based model designed for malware detection. We begin with Section 3.1, which discusses the challenges of utilizing API sequences for malware recognition. Moving on to Section 3.2, we comprehensively describe our model’s architecture, focusing on the transformer framework and the downstream classifier. In Section 3.3, we delve into the preprocessing steps involved in handling API sequences. Lastly, Section 3.4 and Section 3.5 are dedicated to detailing the components of our model, which include the API feature enhancement, the word frequency enhancement module, and the CNN- and LSTM-based downstream classifier.

### 3.1. Challenges

In the process of utilizing API sequences for malware classification tasks, we must consider the following challenges:How to identify obfuscated malware. Malware authors commonly employ various obfuscation techniques within their code to delay malware detection or increase the reverse engineering burden for analysts. These techniques may involve inserting meaningless or scrambled code segments. Consequently, the corresponding API sequences exhibit noise caused by these obfuscation structures. Effectively dealing with this noise in the API sequences becomes crucial in optimizing the model’s performance.How to identify unknown samples. With advancements in detection technology, malware continues to evolve. However, authors often reuse old code segments, leading to similarities in the paths of their API calls [28]. Learning the characteristics of specific API segments aids researchers in identifying attacks originating from the same family.How to focus on key APIs that can determine the maliciousness of a sequence. Malware typically follows a fixed attack route to compromise user devices, resulting in specific APIs being executed multiple times. Paying attention to specific segments of APIs within a sequence helps the model better grasp API semantics, thereby enhancing detection effectiveness.

### 3.2. System Framework

We propose a transformer-based detection model to effectively identify malware API sequence features, as shown in Figure 1. The framework aims to enhance the features of specific APIs in different types of malware for improved performance in classification tasks.

This study uses pre-processed API call sequences as the training corpus after completing the pre-processing. During the API extraction phase, we utilized a publicly available API dataset as the experimental data for our research. To mitigate the impact of noise in the dataset, we initially subjected it to a consistent pre-processing workflow, which will be elaborated in the following chapter.

After completing the pre-processing, we extracted the TF-IDF frequency of each API in the sequence for each dataset sample to be used in weighting operations, enhancing feature representation. Next, we employed an improved transformer model to generate word vector representations for each sample sequence. This step will be detailed in Section 3.4.

During the multi-scale feature extraction stage, we used three one-dimensional convolutions with different scales to extract n-gram features of varying lengths and then concatenated them. We employed a bidirectional LSTM model to learn the semantic correlations among these features. Ultimately, a multi-layer perceptron (MLP) was utilized as the downstream classifier to classify the labels of the target samples.

### 3.3. API Sequence Pre-Processing Method

Eliminating redundant APIs from malware API sequences has proven effective [29,30,31,32]. Our research used the following three commonly used methods to remove duplicate calls.

Single continuous repetition. Malware often exhibits repetitive execution of identical operational behaviors involving various files or requests for multiple resources, as observed in sequences like “regopenkeyexa regopenkeyexa regopenkeyexa ntopenkey ntqueryvaluekey ntclose”, where “regopenkeyexa” is invoked multiple times. Additionally, malicious actions might undergo multiple retry attempts due to unsuccessful implementation. These repeated API calls may indicate preparatory work or preliminary steps the malware takes before engaging in malicious behavior. Concurrently, malware developers may intentionally introduce this continuous repetition to increase the complexity of analysis, confusing researchers and making the malware more deceptive. This makes it challenging for analysts to determine their true intentions and actions accurately.

Paragraph repetition. Malware authors often reuse the same code logic multiple times to expedite malware creation or increase the difficulty of reverse engineering for analysts. For example, in the sequence “NtQueryValueKey LdrUnloadDl RegCloseKey NtQueryValueKey LdrUnloadDl RegCloseKey NtReadFile GetFileSize”, the combination of “NtQueryValueKey LdrUnloadDl RegCloseKey” is repeatedly employed. Additionally, to achieve continuous malicious activities, information gathering, or data transmission, the code may adopt a looping structure, leading to the repeated execution of code blocks within the loop.

Too-short sequences. The extraction results of short API sequences often reveal abnormal interruptions in the execution process of malware, such as the program halting after executing “_CorExeMain”, “_CorDllMain”, or “exception”. This interruption can occur for two reasons: Firstly, malware may employ anti-analysis techniques to disrupt the extraction of behavioral information by dynamic analyzers. When the malware detects a virtualized environment, it actively interrupts its execution process to avoid detection. Secondly, to evade static or dynamic analysis detection, malware may employ strategies such as delayed execution or execution only under specific conditions or environments. This approach conceals its malicious behavior, making detection more challenging.

After removing noise, we set the API sequence length to a fixed value. APIs exceeding this length are truncated, and those with insufficient length are padded with “0”.

### 3.4. API Feature Generation and Enhancement

Word2Vec has achieved significant success in the field of NLP [33]. This word embedding technique is based on shallow neural networks, representing the meaning of a word as a high-dimensional feature vector derived from the distribution of the word in its context. These embedding feature vectors capture semantic and syntactic relationships between words and exhibit semantic similarities suitable for NLP tasks such as word clustering and similarity calculation.

However, the Word2Vec model does not consider the positional information of words in the input sequence; it primarily focuses on the co-occurrence patterns of words, disregarding their order and positional information within sentences. Suppose the training data is of a small scale or lacks representativeness, resulting in less satisfactory generalization of embedding effects. In our research, we employed an innovative approach by utilizing a transformer model for embedding the processing of API sequences. Specifically, we first constructed an API vocabulary based on API tokens in the tracking corpus and represented each API name with a numerical value. Subsequently, we trained a transformer model on the API token sequence to capture the correlations between consecutive API calls.

In this paper, we utilize the transformer model to capture hidden contextual connections between features and effectively model the patterns within API sequences. Our transformer model consists primarily of three parts: an embedding section, an encoder section, and a one-dimensional channel attention enhancement section. As illustrated in Figure 2, the API sequence is input into the encoding layer, composed of six stacked encoders, after undergoing token embedding and positional embedding. We modified the encoder structure of the traditional transformer, and the modified encoder layer comprised multi-head attention layers, a position-wise feedforward network, and a one-dimensional channel attention module. The final one-dimensional channel attention module effectively assists the transformer in expressing the semantic features of APIs.

#### 3.4.1. API Embedding Layers

In past studies, researchers typically used n-gram features of API sequences as the basis of their analysis [10,15,32]. However, our research indicates that employing this method significantly increases the number of tokens due to the complexity of API composition. Using n-grams of API sequences as features results in a huge input dimension for the embedding layer, leading to data sparsity issues and consequently reducing the expressive capacity of the embedding layer. Therefore, learning the semantic relationships and grammatical structures embedded within the data becomes quite challenging.

Before inputting the API sequence into the transformer, it must undergo token embedding and positional encoding operations to map the API names appropriately.

The token embedding function is formed by the fundamental embedding layer, producing vector representations for text sequences. Positional encoding represents temporal features in the data by employing sine and cosine functions to calculate positional embeddings. This enables the model to capture specific locations in sequential data adeptly without explicitly introducing positional information. As a result, the model can understand the relative positional relationships among various positions. Equations (1) and (2) are presented below:(1)$$\mathrm{PosEncoder}\left(\mathrm{pos},\;2i\right)=\;\sin\left(\frac{\mathrm{pos}}{10000^{\left(2i/d_{\mathrm{model}}\right)}}\right)$$
(2)$$\mathrm{PosEncoder}\left(\mathrm{pos},\;2i+1\right)=\;\cos\left(\frac{\mathrm{pos}}{10000^{\left(2i/d_{\mathrm{model}}\right)}}\right)$$

Here, i represents the positional index (starting from 0), pos denotes the dimension index of the positional embedding (also starting from 0), and $d_{\mathrm{model}}$ represents the input dimension of the model.

#### 3.4.2. Encoding Layers

The encoding layer transforms API embeddings into a more comprehensive representation by assigning different weights to different parts, focusing on the essential sections of the input sequence to capture semantic and contextual information. Within the transformer architecture, the encoding layer consists of six iterated encoders.

The incorporation of the self-attention mechanism enables us to more effectively capture long-range dependencies among elements within API sequences. The core process of the self-attention mechanism involves calculating attention weights through Q and K, which are then applied to V to obtain the overall weighted output. This process can be conceptually understood as mapping a query to a series of key-value pairs. Specifically, for input Q, K, and V, the formula for computing the output vector is represented in Equation (3).
(3)$$\mathrm{Attention}\left(Q,\;K,\;V\right)=\mathrm{softmax}\left(\frac{Q\cdot K^{T}}{\sqrt{d_{k}}}\right)\cdot V$$

However, the self-attention mechanism has some drawbacks when encoding information at the current position, and it tends to concentrate attention excessively on its own position [34]. Using a multi-head attention mechanism allows the output of the attention layer to contain encoding representations from different subspaces, thereby enhancing the model’s expressive capability. The calculation method for multi-head attention is shown in Equations (4) and (5).
(4)$${\left\{\mathrm{head}\right\}}_{i}=\mathrm{Attention}\left(Q\cdot {W_{i}}^{Q},K\cdot {W_{i}}^{K},V\cdot {W_{i}}^{V}\right)$$
(5)$$\mathrm{MultiHeadAttention}\left(Q,\;K,\;V\right)=\mathrm{Concat}\left({\left\{\mathrm{head}\right\}}_{1},\cdots ,{\left\{\mathrm{head}\right\}}_{h}\right)\cdot W^{O}$$

Here, Q, K and V represent the query, key, and value vectors respectively. ${W_{i}}^{Q}$, ${W_{i}}^{K}$, and ${W_{i}}^{V}$ are the weight matrices for the i th attention head. $W^{O}$ is the output weight matrix.

#### 3.4.3. One-Dimensional Channel Attention Module

In natural language processing, representing each word with high-dimensional vectors is a widely adopted approach by researchers. Word embeddings map each word to a continuous, high-dimensional real-number vector space, placing words with similar semantics closer together in the vector space. In contrast, semantically unrelated words are positioned farther apart. When API names are embedded into high-dimensional feature vectors, their semantic information is also dispersed across different positions in the vector. The purpose of designing this module is to enhance the model’s understanding of API semantics by capturing features at specific positions.

In their work, Woo et al. [35] presented an effective feed-forward Convolutional Block Attention Module (CBAM) that utilizes conventional operations to extract features from vectors’ maximum and average pooled outputs. This module directs the network’s attention toward relevant elements and spatial locations, emphasizing crucial features while suppressing responses from irrelevant regions. Building on the original CBAM, we extend its capabilities by adapting the channel attention module to focus on essential features within a 1D vector, thus broadening its applicability beyond the image domain.

The modified 1D channel attention module is used at the end of the encoder and iterated through six layers to enhance the feature vector output by the encoder. The specific process is shown in Algorithm 1.
**Algorithm 1** CAFTrans encoder algorithm combining 1D channel attention**Input:** API embedding vector *Src*, padding mask *Src_key_padding_mask*.
**Output:**
*Encoder_output*.
1. Use_1D channel attention ← True
2. *Encoder_output* ← Src
3. for *encoder_layer* ← CAFTrans Encoder ModuleList **do**
4.    *Encoder_output* ← encoder_layer(*Src*, *Src_key_padding_mask*)
5.    **if** Use_1D channel attention is True **then**
6.       *Attention_Weight* ← 1D ChannelAttention Function(*Encoder_output*)
7.       *Encoder_output* ← Vector weighting operation (*Attention_Weight*, *Encoder_output*)
8.       *Encoder_output* ← Norm(*Encoder_output*)
9.    **end if**
10. **end for**
11. **return** *Encoder_output*

Figure 3 shows our proposed one-dimensional channel attention module, which operates as follows: Firstly, it extracts the 1D maximum pooling features and 1D average pooling features of the API feature vectors using nn.AdaptiveMaxPool1d() and nn.AdaptiveAvgPool1d() respectively. Subsequently, these two features are passed into the 1D convolutional feature extraction module, which consists of two convolutional maps of different lengths. The first convolutional network reduces the features’ dimensionality, followed by applying the RELU() activation function. Next, the second convolutional network restores the features to their original dimensions. Lastly, the sigmoid function activates the attention weights, confining them within the range {0, 1}. This process is depicted in Equation (6).
(6)$$\mathrm{CA}\left(x\right)=\sigma \left(\mathrm{MLP}\left(\mathrm{MaxPool}1d\left(x\right)\right)+\mathrm{MLP}\left(\mathrm{AvgPool}1d\left(x\right)\right)\right)\;=\sigma \left(W_{1}\left(W_{0}\left(x_{\max1d}^{c}\right)\right)+W_{1}\left(W_{0}\left(x_{\mathrm{avg}1d}^{c}\right)\right)\right)$$

Here $x_{\max1d}^{c}$ and $x_{\mathrm{avg}1d}^{c}$ represent the max pooling feature and the average pooling feature, respectively. The final channel attention feature map generated in the multi-layer perceptron (MLP) network is $\mathrm{CA}\in R^{C/r\cdot 1}$, and r stands for the number of dimensionality reduction.

Using the channel attention module allows learning the correlation between dissimilar channels and weighting the importance of each channel. The model can enhance its performance and representation capacity by adaptively selecting salient feature channels.

#### 3.4.4. TF-IDF API Frequency Enhancement Module

The TF-IDF technique is the most widely embraced word weighting method in natural language processing. Its widespread applications span a variety of domains, including information retrieval, text mining, modeling, and more. Recently, researchers have extended TF-IDF to API sequence classification tasks [16,19,32], employing this method to assess the significance of individual APIs within the context of the entire sequence. The equations for the calculations are illustrated in Equations (7)–(9)
(7)$${\mathrm{TF}-\mathrm{IDF}}_{i,j}={\mathrm{TF}}_{i,j}\times {\mathrm{IDF}}_{i}$$
(8)$$\mathrm{TF}(i,j)=\frac{n_{i,j}}{\sum_{k}n_{k,j}}$$
(9)$${\mathrm{IDF}}_{i}=\log\frac{\left|D\right|}{\left|\left\{j:t_{i}\epsilon d_{j}\right\}\right|}$$

While *TF-IDF* can reduce the emphasis on high-frequency API calls, it may lose some crucial information for low-frequency calls. Due to the transformer’s self-attention mechanism, generated API sequence word vectors often exhibit longer contextual dependencies and can simultaneously consider global and local semantic information. In *TF-IDF* weighting, low-frequency calls may receive smaller weights, but these calls could indicate the malicious nature of the sequence, information that previous studies might overlook.

To improve *TF-IDF* weights, we introduce a baseline weight α. Subsequently, the weights undergo a normalization process, mapping them to the [0, 1] range, and serve as the final weights. This ensures that the model, while focusing on high-frequency calls, does not overlook low-frequency calls, ensuring that the transformer word vector information can be better perceived by the downstream classifier in capturing the features of the API, as demonstrated in Equation (10).
(10)$${\mathrm{TF}-\mathrm{IDF}}_{i,j}=\mathrm{Norm}({\mathrm{TF}-\mathrm{IDF}}_{i,j}+\alpha )$$

### 3.5. Downstream Classifier

Introducing convolutional neural networks (CNNs) has led to significant research achievements in visual and natural language processing domains. Kim [36] was the first to apply CNNs to text classification tasks, discovering that this method could effectively extract contextual semantic relationships between textual content. Meanwhile, TextCNN, known for its simplicity and speed, has successfully identified local text features. It has been applied in tasks such as text classification, semantic representation learning, and sentiment analysis, demonstrating excellent performance. Consequently, more researchers have begun to explore the potential of TextCNN. Our study employed multiple TextCNN convolutional modules to capture feature relationships of one-dimensional API vectors at different distances.

The fundamental idea of TextCNN is to utilize the structure of CNNs for classification. The model takes a sequence of word vectors and uses different window sizes to capture local information in the sentence and extract relevant features. In our experiments, we employed three different convolutional kernels of sizes 3, 5, and 7 to extract dependencies between different distances in API sequences, the orange part represents the result after processing with a kernel of size 3, the green part represents the result after processing with a kernel of size 5, and the blue part represents the result after processing with a kernel of size 7, as illustrated in Figure 4.

By embedding the API sequence using a transformer, we obtained a matrix with a fixed length of S and a hidden vector length of 512. This matrix is represented as: ${\mathrm{API}\_\mathrm{SEQUENCE}}_{1:S}={\left({\mathrm{API}}_{1},\cdots ,{\mathrm{API}}_{S}\right)}^{T}$.

For further processing, we used this matrix as an input for a CNN with three different kernel sizes. During the convolution process, we slid convolutional kernels of varying window lengths over the vectors represented by ${\mathrm{API}}_{1}$ to ${\mathrm{API}}_{n}$, with a sliding step of 1. When the kernel length is K, a sequence of length S is processed into a sequence of length S - K+1, represented as:${\left({\mathrm{API}}_{1},\;\cdots ,\;{\mathrm{API}}_{S}\right)}^{T}={\;\left(X_{1},\;\cdots ,{\;X}_{S-K+1}\right)}^{T}$. Due to the use of kernels with different sizes, the generated sequence lengths vary. We applied a padding operation to ensure uniform sequence length, adding placeholders on both sides of the vectors. After the convolutional processing, the GELU activation function was employed to better capture information in the negative regions to some extent.

To integrate the crucial features extracted through convolution, we conducted an element-wise summation of the convolutional results$,\;X\;=\mathrm{LayerNorm}\left[X_{3}⊕X_{5}⊕X_{7}\right]$, where *X* represents the fused features, and *LayerNorm* denotes the layer normalization operation. The *LayerNorm* operation was employed to standardize the features of each sample, promoting a smoother feature distribution across various samples to enhance the model’s generalization capability [37]. Additionally, it aids in mitigating challenges like vanishing and exploding gradients, expedites convergence, and augments the overall generalization capacity of the model.

Due to the unique syntax of API names in classifying malware API sequences, they can also be considered a particular type of text. By storing long-distance information in the hidden state and short-term memory in the cell state, the sequential dependencies can be captured to provide richer semantic representations. Numerous researchers have demonstrated through experiments [18,38] that the application of long short-term memory (LSTM) [39] networks can drive the development of malware detection and classification techniques.

Following the concatenation of all convolutional layer outputs, the Bi-LSTM network processes the data in both forward and reverse directions, facilitating the capture of relationships between API calls.

Due to the unique gate structure of LSTM networks [40], they can selectively learn information in API sequences, thereby capturing long-term dependencies in the sequences. Our experiments employed Bi-LSTM, which considers past and future features, unlike traditional unidirectional LSTM. It utilizes two LSTM layers, one to process the input in the original sequence order and one in the reverse sequence order. The outputs from both LSTM layers are then concatenated to capture bidirectional semantic relations between the APIs. This can be expressed as Equations (11)–(13).
(11)$$\vec{h_{t}}=\vec{\mathrm{LSTM}}\left(h_{t-1},{\;W}_{t},\;C_{t-1}\right)$$
(12)$$\overset{⃐}{h_{t}}=\overset{⃐}{\mathrm{LSTM}}\left(h_{t-1},\;W_{t},\;C_{t-1}\right)$$
(13)$$H_{t}=\left[\overset{⃐}{h_{t}},\;\vec{h_{t}}\right]$$

Here $\vec{h_{t}}$ represents the forward feature information, and $\overset{⃐}{h_{t}}$ represents the backward feature information.

While previous researchers often preferred using max-pooling to downsample the final features of the LSTM network [17,41], our experiments have adopted a distinct approach. However, reducing the model’s parameter size causes the loss of temporal order relationships and finer-grained information between time steps, ultimately leading to a decrease in the model’s recognition performance.

Afterward, two linear layers with dimensions (512, 64) and (64, *n*) are used for decision-making, where n represents the number of labels in the dataset. Each linear layer uses GELU activation to output the estimated malware probability. In the experiment, we set the hidden size to 256 and employed AdamW as the optimizer. We used the cross-entropy loss function to measure the loss during the training phase to compute the discrepancy between the label values and the MLP output results.

## 4. Experimental Evaluation

### 4.1. Testing Dataset

Catak et al. [42] detected Windows malware using Cuckoo Sandbox V2.0.6 and labeled the collected malware samples using the online service VirusTotal. They released a publicly available benchmark dataset named ‘mal-api-2019’. Malware from different families in the dataset can be classified into eight classes: spyware, downloaders, trojans, worms, adware, droppers, viruses, and backdoor malware. The dataset comprises 7107 API call sequences, including 342 different Windows system calls. The distribution of samples for each category is shown in Table 1 below.

In this paper, we equally sample different types of software and employ four standard evaluation metrics—accuracy, recall, precision, and F1 score—as well as an ROC curve and the area under the curve (AUC) to assess the performance of CAFTrans, as formulated in Equations (14)–(17).

Additionally, we rely on the false positive rate (FPR) and false negative rate (FNR) to assess the model’s discriminative ability for different types of samples. The false positive rate (FPR) represents the proportion of all actual negative samples incorrectly predicted as positive by the model. In contrast, the false negative rate (FNR) represents the proportion of all actual positive samples incorrectly predicted as negative by the model, as formulated in Equations (18) and (19).
(14)$$\mathrm{Precision}=\frac{\mathrm{TP}}{\mathrm{TP}+\mathrm{FP}}$$
(15)$$\mathrm{Recall}=\frac{\mathrm{TP}}{\mathrm{TP}+\mathrm{FN}}$$
(16)$$\mathrm{ACC}=\frac{\mathrm{TP}+\mathrm{TN}}{\mathrm{TP}+\mathrm{TN}+\mathrm{FP}+\mathrm{FN}}$$
(17)$$F1-\mathrm{score}=\frac{2\;\times \;\mathrm{Precision}\;\times \;\mathrm{Recall}}{\mathrm{Precision}+\mathrm{Recall}}$$
(18)$$\mathrm{False}\;\mathrm{Positive}\;\mathrm{Rate}(\mathrm{FPR})=\frac{\mathrm{FP}}{\mathrm{FP}+\mathrm{TN}}$$
(19)$$\mathrm{False}\;\mathrm{Negative}\;\mathrm{Rate}(\mathrm{FNR})=\frac{\mathrm{FN}}{\mathrm{FN}+\mathrm{TP}}$$

### 4.2. Experimental Setup

The experiment used a local server with the following configuration: CPU: Intel i9-11900K @ 3.50 GHz, GPU: NVIDIA RTX 3090, 128 GB RAM. The code was developed on a Windows 10 platform, built using PyTorch version 1.13, and PyCharm was used as the Python compiler with Python 3.10 as the programming language environment.

To partition the training and testing sets, we initially classified malware calls based on the different labels present in the dataset. Subsequently, we used an 8:2 ratio to achieve a balanced collection of API call data for each label, ensuring a consistent representation for each category. This approach enhanced the generalization ability of the model, minimized the bias effects, reduced the risk of overfitting, and ultimately improved the model’s stability.

In order to expedite the detection of malicious behavior during the experiments, we standardized the API sequence length to accept 200 calls. Our network training utilized a learning rate of 1 × 10^−4^, a maximum of 100 epochs, and a batch size of 64. Table 2 lists the parameters used in the experiments.

Choosing the appropriate number of epochs is crucial in preventing model overfitting. Our experiments observed that when the training epochs exceeded 100, the model’s loss continued to increase, and simultaneously, the accuracy gradually fluctuated. This indicates that the model was overfitting to the data. We achieved the best experimental results during the model training process at the 89th epoch. We set the maximum training epoch to 100 to prevent overfitting due to excessive training.

Choosing the right batch size is equally crucial for achieving optimal results. A too-large batch size can lead to increased training time and memory overhead, while a too-small batch size may result in insufficient data generalization. To strike a balance between accuracy during detection and the associated training time and memory costs, we opted for a batch size of 64.

### 4.3. Evaluation

In this section, we present the experimental results of the model, evaluate its performance improvements through a comparison with baseline models, conduct ablation studies, conduct data cluster analysis, and test against unknown samples. Additionally, we assess its enhanced generalization capability for recognizing different types of samples.

#### 4.3.1. Baselines Comparative Evaluation

To validate the superiority of our proposed model, we compared its performance with several baseline models. These baseline models include GaussianNB, LogisticRegression, KNeighbors, SVM, DecisionTree, and RandomForest, as shown in Table 3. Simultaneously, we gathered studies on the mal-api-2019 dataset, encompassing experimental results from five researchers. We compiled and analyzed these results as reference data, and the comparative outcomes are illustrated in Table 4.

From Table 4, Demirkıran et al. [43] tested multiple pre-trained models, including BERT, CA-NINE-S, and their proposed Random Transformer Forest (RTF) model, on the mal-api-2019 dataset. Li et al. [44] utilized RNNs and transformer structures to classify malware by learning interactive features in API call sequences. Catak et al. [45] evaluated the effectiveness of different LSTM network structures by using individual API numbering as model inputs on the mal-api-2019 dataset. Avci et al. [46] assessed and benchmarked LSTM-based malware detection methods on specific LSTM architectures, showing that different LSTM approaches and architectures applied to the malware detection problem. Cannarile et al. [47] explored the ability of various tree-based methods and recurrent neural networks to capture patterns in API relationships.

Compared to other researchers’ studies, our proposed model exhibits a lower false positive rate, achieving a precision value of 0.65140, surpassing the results of Cannarile et al. by 5%. Additionally, it demonstrated a lower false negative rate, achieving a recall value of 0.65842, outperforming the results of Cannarile et al. by 8%. The model’s high recall rate indicates that we effectively reduced false negatives during detection, better identified correct labels for test samples, and minimized the number of false negatives (missed true positives), which is crucial in malware detection. Compared to all baseline algorithms, our proposed model demonstrates superior overall performance with an F1 score of 0.65252, outperforming the best result achieved by Demirkıran et al. with 0.6149.

Figure 5 shows the confusion matrix of our proposed model for malware classification on the test dataset. It shows that the model performs best in identifying malware types as adware, downloaders, and viruses, with accuracy values of 80%, 72.35%, and 80.189%, respectively. For other types of samples, the model achieves a precision rate of more than 50%, demonstrating its effectiveness in identifying various forms of malware. However, the model performs poorly in identifying trojan samples, as they are more prone to misclassification as backdoor software. One possible explanation for this observation is that the creators of trojans may deliberately exploit system vulnerabilities to install backdoors and gain system access privileges.

#### 4.3.2. Ablation Studies

In this study, we propose three significant enhancements to our model architecture to improve feature recognition. First, we incorporate a joint CNN+LSTM classifier, which leverages this composite model in downstream tasks to enable feature learning at varying distances. Second, we refine the TF-IDF call strategy to enhance its role within the model. Finally, we introduce a 1D channel attention module to augment the embedding features. To assess the impact of each improvement on feature recognition, we conducted ablation experiments by systematically turning off specific modules and measuring their effect on the model performance. Throughout the experiment, we integrated the transformer model with the downstream CNN+LSTM classifier to create the primary base model MTran.

As illustrated in Table 5, we conducted experiments on the mal-api-2019 dataset by selectively deactivating specific modules and evaluating performance metrics, including ACC, Precision, Recall, F1-score, and AUC (macro). As modules were added, the model’s performance consistently improved, as evidenced by the incremental expansion of the ROC curve area. Figure 6 shows that the original transformer model achieved an ROC area of 0.88383, while our proposed enhanced model achieved an impressive ROC area of 0.89125, indicating a notable gain of 0.742%. Additionally, the ACC increased from 62.299% to 64.611%, representing a significant improvement of 2.3%. The ROC plot demonstrates that including any intrinsic feature extraction module in our proposed model leads to a performance enhancement, resulting in an increased AUC score. This observation indicates that each designed intrinsic feature extraction module effectively contributes to malware detection, enabling the model to adapt to intricate detection scenarios. Consequently, our work is substantiated in terms of its valuable contribution to the model’s classification performance, resulting in improved prediction accuracy and robustness.

Next, we tested the impact of the TF-IDF module and the 1D channel attention module on the performance of the transformer model in malware identification. Enabling the TF-IDF module alone yielded an F1-score of 0.64259, while enabling the 1D channel attention module alone yielded an F1-score of 0.64167. Both individually led to a slight performance improvement compared to the scenario without these modules. More importantly, however, when both modules were enabled, the F1-score increased from 0.62397 to 0.65252, showing a significant improvement of 3%. These results indicate that combining TF-IDF and 1D channel attention modules significantly enhances the model’s ability to learn API semantic relations.

The downstream classifier proposed in the study, which combines a multiscale convolutional network with an LSTM, significantly improves the detection performance of the model. In the experiments, we compared the accuracy, F1 score, and AUC score of the original transformer model with our proposed transformer model, incorporating the downstream classifier, using the mal-api-2019 dataset. The results demonstrate that the accuracy, F1 score, and AUC score of the original transformer model were 0.62299, 0.62397, and 0.88383, respectively. In contrast, the transformer model with the added downstream classifier showed performance improvements, achieving an accuracy of 0.63139, an F1 score of 0.63552, and an AUC score of 0.88456.

By combining these techniques, our research enables flexible learning of information within sequences and captures long-term dependencies in API sequence analysis tasks. The proposed feature-capturing module significantly enhances the performance of transformer models, particularly in handling complex API semantic relationships. These results provide valuable guidance for API sequence classification tasks.

#### 4.3.3. Data Cluster Analysis

To demonstrate the model’s classification performance more clearly, we employed T-SNE plots [48] for data visualization to better understand and illustrate the relationships between samples. First, we used the PAC algorithm to map the 3D features of samples in the original feature space to a 2D space. Subsequently, we compared the original transformer model with the proposed model in this study. We presented T-SNE plots for the testing samples of malware detection and classification tasks (Figure 7a,b).

By comparing the two plots, we observed that the samples in our proposed model tended to cluster more clearly and densely. In contrast, the samples in the T-SNE plot of the original transformer model were more scattered and showed higher overlap among samples with different labels. This indicates that the original transformer model struggles to effectively distinguish these samples, whereas our method can better capture crucial features within the malware API sequence and more effectively identify different types of malware.

#### 4.3.4. Unknown Sample Attack Detection

As the expertise of malware developers continues to advance, a decisive factor in determining the effectiveness of a model is its ability to detect unknown samples. Our study tested the model’s efficacy in detecting unknown sample attacks and compared it with an LSTM model, obtaining satisfactory results.

We uniformly extracted 5% of unknown samples from various malware categories in mal-api-2019, with 80% as the training set and 15% as the validation set. We evaluated the training performance of the model over 100 epochs and compared its performance on the combined dataset of unknown samples and the validation set (20% of the dataset), simulating real-world scenarios of facing unknown sample attacks. During training, we reduced the number of training samples, leading to a decrease in identification accuracy. However, this decrease is deemed negligible. In this section, our primary focus is on the disparity in detection performance.

By carrying out performance comparisons on the combined dataset (unknown samples) and validation set (original samples), we can simulate scenarios of facing unknown sample attacks in the real world. This is crucial as the expertise of malware developers continues to advance, making the model’s capability to detect unknown samples increasingly pivotal [28]. The comparative results, as shown in Table 6, reveal that our model experienced a minor decrease of 0.28% in accuracy and 0.16% in F1 score when tested on unknown samples. In contrast, the LSTM model exhibited a more substantial decline, with a 9.99% decrease in accuracy and a 10.51% decrease in F1 score. These findings indicate that our model demonstrates superior robustness in the face of attacks involving unknown samples in scientific research.

Our experimental findings reveal the model’s resilience to unknown samples within the same family type. Malware from the same family typically shares standard features. From our experiments, this model can effectively extract semantic features from the API sequences of the same family of malware, which is crucial for distinguishing between different types of malware. However, this study did not undertake the task of detecting unknown categories of malware. Typically, this requires the collection of new samples for training. Nevertheless, these anonymous malware categories often demonstrate a high similarity in API calls to existing samples, leading us to believe that our model can also handle unknown malware categories.

For entirely new unknown categories of malware, the API sequences they need to execute often exhibit high similarity in specific segments to existing samples. This is because, even though they are new samples, their operational behavior on the system may follow patterns. By leveraging the API semantic relationships extracted by our model, this model can also confront entirely new unknown categories of malware samples.

## 5. Limitations and Future Work

### 5.1. Additional API Parameter

In this study, we have only used the dataset containing API names and have not incorporated other information, such as parameters and execution time intervals. However, recent research has revealed the importance of these additional details in uncovering crucial malware features [20,49]. Still, extracting this information using virtual machines and sandboxes is a time-consuming process. Therefore, in the future, we plan to integrate API parameters and related knowledge from the security domain to design a more specialized and comprehensive API sequence detection model. Considering such information, the model will be better equipped to analyze malware and provide a more comprehensive security assessment accurately. This approach will lead to improved detection capabilities and a more thorough evaluation of security aspects.

### 5.2. Concept Drift Problem

Concept drift refers to the problem of changing underlying relationships in the data. Concept drift poses a significant challenge in the context of malware detection tasks. As the techniques used by malware authors continually advance, new types of malware become increasingly challenging to detect, leading to a decline in the prediction quality of malware detectors and classifiers over time [50]. To address concept drift, we will integrate security knowledge related to API calls and explore more efficient approaches that can continuously detect new types of malware.

### 5.3. Malicious Paragraph Localization Issue

In our experiments, we used only the first 200 API calls as features to identify different API sequences quickly. However, since malware may lurk in a user’s system for an extended period to avoid early detection, this could result in malicious behavioral features appearing in the middle or at the end of the sequence. Therefore, the practice of detecting only the initial API calls in this experiment may have some limitations in specific scenarios. Using graph structures allows for a complete representation of all parts of the API sequence, enabling the graph model to capture the complexity of malware behavior more comprehensively. In subsequent experiments, we will test the effectiveness of graph structures in the classification of malware API sequences, aiming to handle the intricate and variable patterns of malware behavior.

## 6. Conclusions

In this study, we have proposed an innovative transformer architecture named CAFTrans to effectively learn the intrinsic features of diverse types of malware API sequences. We have designed a novel transformer structure to achieve improved API embedding effects, leading to more accurate identification of various types of attacks. The API sequences undergo two rounds of feature enhancement, including one-dimensional channel attention reinforcement and TF-IDF API Frequency Enhancement, before inputting into the final CNN-LSTM classifier for decision-making.

We have compared our model with various machine learning algorithms and other researchers, and the results demonstrate the outstanding performance of our architecture in malware detection tasks. On the mal-api-2019 dataset, we achieved an F1 score of 0.65252 and an AUC score of 0.89463. Furthermore, our architecture exhibits robustness and generalization in detecting unknown samples, effectively learning the semantic information of complex API sequences and characteristics of different types of attacks. We have confidence in our proposed transformer model, believing that it can assist analysts in rapidly and accurately identifying different malware families to counter evolving attack methods.

## Figures and Tables

**Figure 1 sensors-24-00580-f001:**
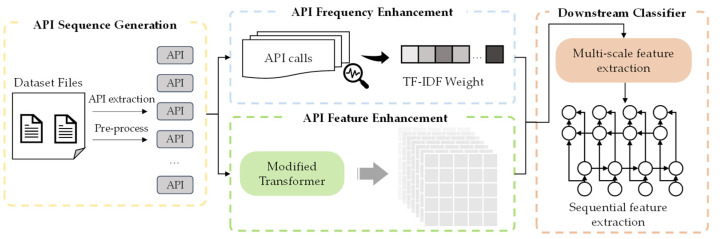
The proposed CAFTrans system framework.

**Figure 2 sensors-24-00580-f002:**
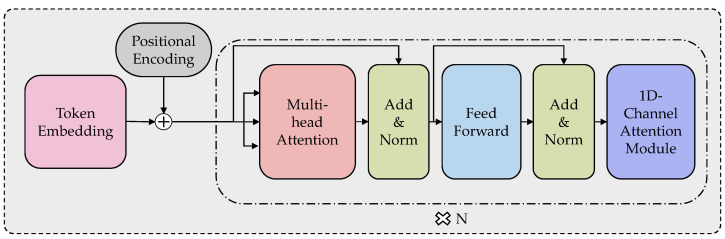
Modified transformer with 1D-channel attention module.

**Figure 3 sensors-24-00580-f003:**
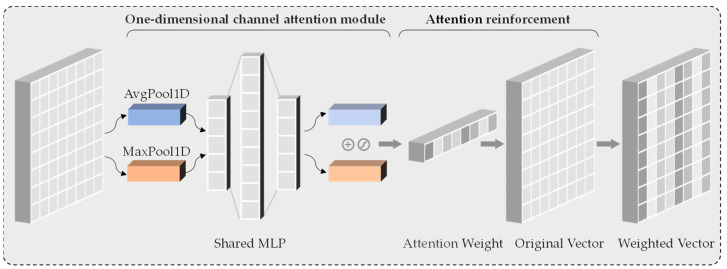
The enhancement process of one-dimensional channel attention to transformer output vector.

**Figure 4 sensors-24-00580-f004:**
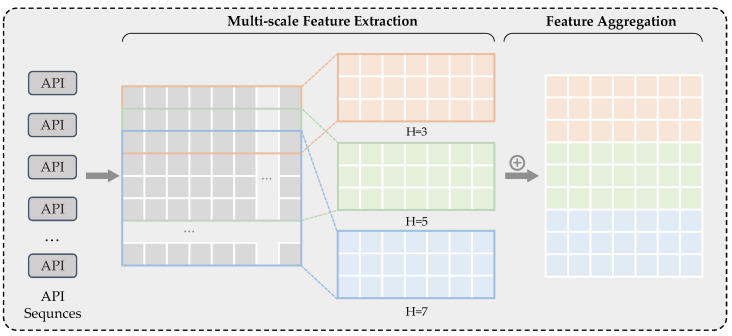
Multi-scale feature extraction using convolutional networks with different convolution kernels.

**Figure 5 sensors-24-00580-f005:**
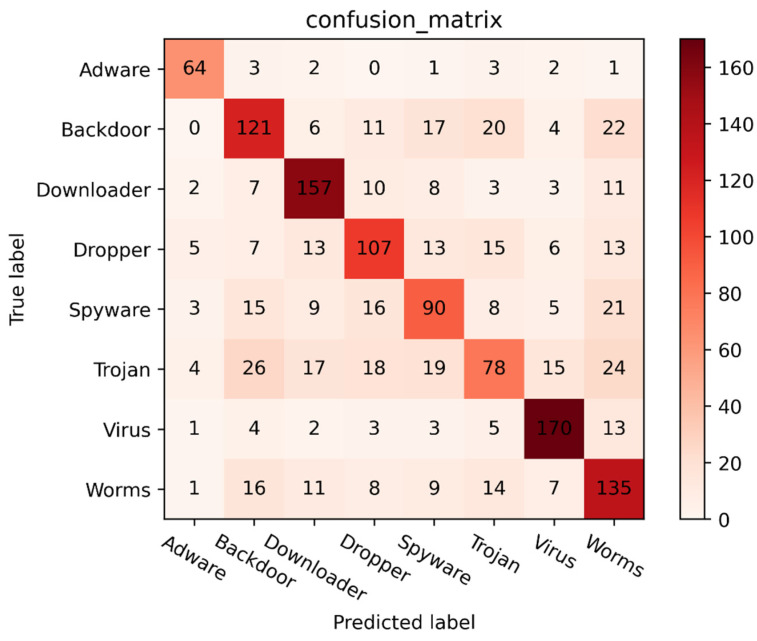
Confusion matrices obtained by CAFTrans on mal-api-2019 dataset.

**Figure 6 sensors-24-00580-f006:**
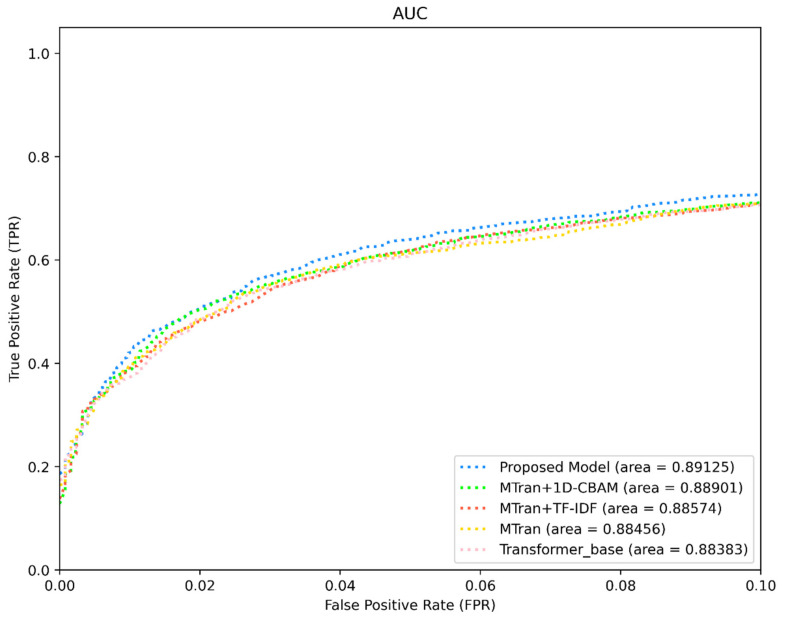
Comparison of ROC curves of each module of CAFTrans.

**Figure 7 sensors-24-00580-f007:**
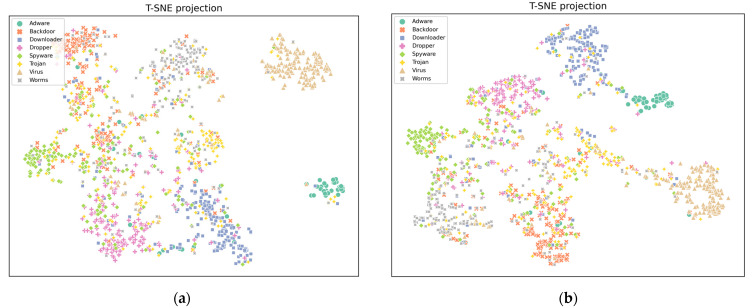
Comparison of clustering results. (**a**) T-SNE projection of the original transformer clustering effect; (**b**) T-SNE projection of the CAFTrans clustering effect.

**Table 1 sensors-24-00580-t001:** Sample distribution of the mal-api-2019 dataset.

Type	Instance
Worms	1001
Virus	1001
Trojans	1001
Downloaders	1001
Backdoors	1001
Droppers	891
Spyware	832
Adware	379

**Table 2 sensors-24-00580-t002:** Parameter setting in the experiment.

Parameters	Set Value
Optimizer	AdamW
Learning Rate	1 × 10^−4^
Decay	None
Sequence Length	200
Batch Size	64
Epoch	100

**Table 3 sensors-24-00580-t003:** Comparisons with baseline models.

Method	ACC	Precision	Recall	F1-Score	Processing Time (s)
GaussianNB	0.1500	0.2980	0.1891	0.1122	0.016
LogisticRegression	0.2950	0.3138	0.3094	0.3065	0.154
KNeighbors	0.3525	0.4005	0.3770	0.3745	0.006
SVM	0.3665	0.4302	0.37404	0.3868	2.145
DecisionTree	0.4338	0.4494	0.4511	0.4500	0.339
RandomForest	0.5088	0.5456	0.5189	0.5261	1.807
MLP	0.2256	0.3524	0.2127	0.1892	4.892
GradientBoosting	0.4926	0.5173	0.5061	0.5085	48.220
CAFTrans	0.6461	0.6514	0.6584	0.6525	1.721

**Table 4 sensors-24-00580-t004:** Comparisons with other researchers.

Study	Method	ACC	Precision	Recall	F1-Score	AUC
Demirkıran et al. [43]	Random Transformer Forest (RTF)	-	-	-	0.6149	0.8818
	BERT	-	-	-	0.5919	0.8735
	CANINE-S	-	-	-	0.5633	0.8339
Li et al. [44]	Transformer	0.50	0.50	0.52	0.51	-
Catak et al. [45]	Single LSTM	-	0.50	0.47	0.47	-
	Bidirectional LSTM	-	0.40	0.41	0.39	-
Avci et al. [46]	CNN LSTM	0.8847	0.5441	0.1703	0.2483	0.836
Cannarile et al. [47]	ExtraTrees	0.557	0.593	0.571	0.578	0.753
Proposed Model	CAFTrans	0.6461	0.6514	0.6584	0.6525	0.8913

**Table 5 sensors-24-00580-t005:** The ablation experimental results.

Method	ACC	Precision	Recall	F1-Score	AUC
Transformer-base	0.6230	0.6261	0.6332	0.6240	0.8838
MTran	0.6314	0.6321	0.6440	0.6355	0.8847
MTran+TF-IDF	0.6335	0.6455	0.6430	0.6426	0.8857
MTran+1D CAM	0.6321	0.6342	0.6372	0.6417	0.8890
CAFTrans	0.6461	0.6514	0.6584	0.6525	0.8913

**Table 6 sensors-24-00580-t006:** Detection performance of unknown samples and original samples.

Method	Data	ACC	Precision	Recall	F1-Score	AUC
LSTM	Original samples	0.5098	0.5543	0.5543	0.5220	0.8360
	Unknown Samples	0.4099	0.4609	0.4055	0.4169	0.7769
Proposed Model	Original samples	0.6424	0.6453	0.6536	0.6464	0.8961
	Unknown Samples	0.6396	0.6436	0.6484	0.6448	0.8927

## Data Availability

We used the mal-api-2019 dataset in this research. The dataset is available at https://github.com/ocatak/malware_api_class, accessed on 26 November 2023.

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
