# Peer review of "Channel Features and API Frequency-Based Transformer Model for Malware Identification"

_sensors, 2024, doi:10.3390/s24020580_

Round 1

Reviewer 1 Report

Comments and Suggestions for Authors

This research paper presents the CAFTrans model, an innovative approach in the realm of malware detection, leveraging API sequences. The model ingeniously integrates natural language processing (NLP) techniques with a focus on API call frequencies and unique behaviors, marking a significant step forward in cybersecurity technology. The use of a one-dimensional channel attention module and a word frequency reinforcement module within a Transformer network framework is particularly commendable. This combination enhances the distinctiveness of API feature vectors and improves the model's ability to differentiate between various types of malware effectively.

However, there are areas where the article could benefit from improvements, particularly in providing a clearer context for a broader audience:

1.    Remark 1. While the technical aspects of the model are well-explained, the article could benefit from a more detailed introduction to the non-expert reader about the significance of API sequences in malware detection. A brief overview of how API behaviors differ in normal versus malicious software could provide valuable context and make the content more accessible to readers unfamiliar with the subject.

2.    Remark 2. The article assumes a level of pre-existing knowledge about NLP and its application in malware detection, which might not be the case for all readers. A concise explanation of how NLP techniques are traditionally used and how they are being adapted in this new context would enhance understanding and appreciation of the model's innovative approach.

In summary, the CAFTrans model represents a sophisticated and promising advancement in malware detection. Its successful integration of NLP techniques with traditional malware detection methodologies is a notable achievement. However, the article would benefit from more accessible explanations to ensure that the significance and innovation of the model are fully appreciated by a wider audience.

Author Response

Dear Reviewer, we have completed the revisions. For specific details, please refer to the submitted Word document.

Reviewer 2 Report

Comments and Suggestions for Authors

I reviewed the manuscript entitled “Channel Features and API Frequency-Based Transformer Model for Malware Identification” in detail. In this article, the authors proposed a Transformer-based model for malware detection. I found a lot of flaws in the paper that should be fixed to meet the high standards of the Journal as well as the research community. The presented work is good. However, some concerns need to be resolved, in the next revision, which is given as follows: 

Minor revision:

1.      A lot of grammatical mistakes throughout the manuscript should be removed, and scientific language needs to be used to meet the higher standard of the journal.  A lot of grammatical, spelling mistakes, and space issues throughout the manuscript.

2.      In the introduction part, need to explain the limitations of the current research work significantly/separately, as described proposed work limitations in section 5.

3.      What are the main reasons to set the value of epoch 100 maximum and the batch size 64?

4.      Cite and explain all variables that were used in the equations properly. Citation of eq.4-5, 7-9, etc. are missing.

5.      Citations were used somewhere like Fig./Fugure, and Cite figures/tables throughout the manuscript with the same style.

6.      The reference style throughout the manuscript is not the same, somewhere authors used the Number style and somewhere APA style.

7.      The manuscript should be revised by an English native speaker, for the improvement of the sentence structure and story of the paper.

Comments on the Quality of English Language

I reviewed the manuscript entitled “Channel Features and API Frequency-Based Transformer Model for Malware Identification” in detail. In this article, the authors proposed a Transformer-based model for malware detection. I found a lot of flaws in the paper that should be fixed to meet the high standards of the Journal as well as the research community. The presented work is good. However, some concerns need to be resolved, in the next revision, which is given as follows: 

Minor revision:

1.      A lot of grammatical mistakes throughout the manuscript should be removed, and scientific language needs to be used to meet the higher standard of the journal.  A lot of grammatical, spelling mistakes, and space issues throughout the manuscript.

2.      In the introduction part, need to explain the limitations of the current research work significantly/separately, as described proposed work limitations in section 5.

3.      What are the main reasons to set the value of epoch 100 maximum and the batch size 64?

4.      Cite and explain all variables that were used in the equations properly. Citation of eq.4-5, 7-9, etc. are missing.

5.      Citations were used somewhere like Fig./Fugure, and Cite figures/tables throughout the manuscript with the same style.

6.      The reference style throughout the manuscript is not the same, somewhere authors used the Number style and somewhere APA style.

7.      The manuscript should be revised by an English native speaker, for the improvement of the sentence structure and story of the paper.

Author Response

(The authors gave the same response as above.)

Reviewer 3 Report

Comments and Suggestions for Authors

This manuscript presents a malware detection model, CAFTrans, based on API sequences. The model enhances the representation of API features by utilizing natural language processing techniques combined with call frequency and attention to unique API behaviors. The CAFTrans has two distinctive features: a unique one-dimensional channel attention module and a feature reinforcement module based on word frequency. Firstly, the API sequences undergo embedding representation through a transformer network. In each encoder,  a one-dimensional channel attention is employed to reinforce the specificity of features, making API vector features more distinct. Subsequently, a word frequency reinforcement module is applied to enhance the feature values of high-frequency API calls while preserving the features of low-frequency API calls. Finally, a downstream classifier composed of the CNN and LSTM is used to identify the unique correlations in the API sequences of different malware types, enabling better differentiation among various types of malware. The implementation results are also presented that show the advantages of the proposed work. The talked issue is interesting for reader. In addition, the structure is acceptable. This paper can be accepted after minor editing of English language.

Comments on the Quality of English Language

Minor editing of English language is required

Author Response

(The authors gave the same response as above.)

Reviewer 4 Report

Comments and Suggestions for Authors

An improved scheme for API frequency-based malware detection is reported in this article.

The idea of the incorporation of NLP technique into malware identification of its novelty.

The paper is well structured and well presented with good linguistic quality.

Is the proposed scheme applicable to graph-based malware detection?

How the proposed approach is capable of dealing with unknown attacks should be further elaborated.

The proposed method outperforms existing ones. However, there is still a significant gap to perfection. Please address the challenges and possible remedy.

Author Response

(The authors gave the same response as above.)
